# Hydrothermal alteration of Ryugu from a disruptive impact recorded in a returned sample

Devin L. Schrader [1,4] ✉, Thomas J. Zega[2], Maizey C. Benner[2] & Jemma Davidson[3]

Samples returned from C-type asteroid (162173) Ryugu by JAXA's Hayabusa2 spacecraft provide material free of terrestrial alteration for analysis. Previous work shows that Ryugu was aqueously altered below 100 °C under neutral to alkaline fluid conditions. Such low-temperature alteration was unexpected based on spacecraft observations which indicated that Ryugu's surface may be thermally altered. Here we show that sulfides in a single particle (A0016) returned by the Hayabusa2 mission are unlike any previously described from Ryugu. The presence and compositions of violarite, pyrite, chalcopyrite, pentlandite, and Fe-depleted pyrrhotite grains provide direct evidence for hydrothermal alteration between 230 to 400 °C under highly oxidizing and acidic fluid conditions. We hypothesize that A0016 was near to the site of a large impact that disrupted Ryugu's precursor parent body.

The Hayabusa2 spacecraft successfully collected over 5 g of material from the ~0.5 km diameter asteroid Ryugu at two distinct sites, touchdown site 1 (TD1; chamber A samples) and TD2 (chamber C samples), e.g.,[1–3]. The samples from TD1 are surface material, whereas those from TD2 are primarily subsurface material excavated during the formation of the artificial crater made by the Small Carry-on impactor[1,2]. Despite the two collection sites, both sets of Ryugu samples were found to be very similar and consist of material that was extensively aqueously altered at temperatures below 100 °C[3–9].

The near-infrared-spectrometer observations during the orbital phase of Hayabusa2 identified a weak 2.72 μm hydroxyl (-OH) band on asteroid Ryugu, indicating that the surface was thermally altered and relatively dehydrated compared to a more OH⁻-enriched subsurface exposed by the Small Carry-on impactor[10–12]. This observation led to predictions that Ryugu's surface and the returned samples would be thermally altered[10,11,13]. Surprisingly, an analysis of particles from TD1 and TD2 concluded that Ryugu was aqueously altered at temperatures as low as ~30 ± 10 °C (from magnetite-dolomite thermometry[5]), which agrees with the pentlandite-equilibration temperature of at least 25 °C[3]. Such alteration occurred in neutral to alkaline fluid conditions

(pH>8)[3]. After low-temperature aqueous alteration occurred, the Ryugu samples analyzed did not experience peak temperatures above ~100 °C[5].

Ryugu's precursor parent body accreted 1.8 to 2.9 million years after the formation of the first Solar System solids ($t_0$)[3]. Radioactive decay melted water ice at ~3 million years after $t_0$, reaching a peak internal temperature of ~75 °C at 5 million years after $t_0$ (assuming a modeled precursor body ~50 km in radius[3]). About 1 billion years ago, Ryugu's precursor parent asteroid was then disrupted by a large impact that heated material near the impact site up to 700 °C. Fragments from this disruption event, that did not get above 100 °C, reaccreted to form Ryugu[3,14].

Ryugu samples were found to be mineralogically, chemically, and isotopically similar to the intensely altered Ivuna-like carbonaceous (CI) chondrites and described as CI-like material, e.g.,[2,3]. Some Ryugu samples contain silicates (olivine and/or pyroxene) and are considered less aqueously altered than those that do not contain silicates[3]. The CI chondrites are inferred to have undergone low-temperature aqueous alteration below 100 to 135 °C, and potentially as low as 25 °C, based in part on the sulfide mineralogy[15–17]. Recent analyses of samples returned from asteroid Bennu are noted to be similar to both Ryugu and the CI-

[1]Buseck Center for Meteorite Studies, School of Earth and Space Exploration, Arizona State University, Tempe, AZ, USA. [2]Lunar and Planetary Laboratory, University of Arizona, Tucson, AZ, USA. [3]Astromaterials Research and Exploration Science (ARES) Division, XI2 Astromaterials Acquisition and Curation Office, NASA Johnson Space Center, Houston, TX, USA. [4]Present address: Astromaterials Research and Exploration Science (ARES) Division, XI3 Research Office, NASA Johnson Space Center, Houston, TX, USA. ✉e-mail: devin.l.schrader@nasa.gov

chondrites, and have also undergone low-temperature aqueous alteration as low as 25 °C (e.g.,[18]). Sulfides are highly sensitive indicators of aqueous alteration and thermal alteration conditions, such as pH, oxygen fugacity ($fO_2$), and temperature[3,16,19–21]. The sulfides commonly identified in Ryugu are identical to the sulfide minerals identified in CI chondrites; which are dominantly Fe-depleted pyrrhotite [ideally $Fe_{1-x}S$] and pentlandite [$(Fe,Ni)_9S_8$], with minor cubanite [$CuFe_2S_3$] and sphalerite [$(Zn,Fe)S$][2,3,9,15–17,21,22]. At least one instance of digenite ($Cu_9S_5$) and djerfisherite ($K_6Na(Fe,Cu,Ni)_{25}S_{26}Cl$) were also identified in Ryugu[3,23].

The disparity between the heated surface of Ryugu and the low-temperature aqueously altered material in the returned samples could be explained by a thin space-weathered crust covering a hydrated interior[24]. This thin, heated, and potentially dehydrated, crust may have also formed due to solar heating[11]. However, in addition to these explanations, it is possible that extensively heated material from Ryugu exists but (1) it has not yet been analyzed in the returned samples or (2) sampling preferentially collected subsurface material and not the heated crustal material. Here we report on the morphology and chemical compositions of sulfides in three Hayabusa2 particles to bridge this knowledge gap.

## Results

### Sample characterization

We conducted high-resolution X-ray element mapping to determine the location and general elemental compositions of sulfides, high-resolution imaging of sulfides, and quantitative in situ major and minor element analyses of sulfides in three different Ryugu particles (A0016, A0094-01, and C0103-01) using an electron probe microanalyzer (Methods). The Chamber A samples A0016 and A0094 from TD1 are assumed to be surface material. Chamber C sample C0103 is from TD2 and is assumed to be subsurface material excavated due to the impact of the Small Carry-on impactor[1,2]. However, with the sampling mechanism disturbing the asteroid's surface[25], it is possible all samples collected are subsurface samples.

### Sulfides in Ryugu particle A0016

The sulfides identified in A0016 are unexpected, based on what was previously known about sulfides from Ryugu. A0016 contains 4.1 ± 0.1 vol.% sulfides (±1σ), which are dominantly (~4 vol.%) Fe-depleted pyrrhotite, rare (0.07 ± 0.02 vol.%) violarite [ideally, $FeNi_2S_4$], and trace (<0.01 vol%) pentlandite, pyrite [$FeS_2$], and chalcopyrite [$CuFeS_2$] (Table 1 and Figs. 1a and 2). The sulfides are up to ~70 μm in longest dimension (Fig. 1a). One pyrrhotite analysis in A0016 shows a high Ni content (11.2 wt%), which indicates the presence of pentlandite (Data Availability). A single grain of pyrrhotite intergrown with pentlandite was found along the edge of A0016 (Figs. 1a and 2f), but this grain is in the same lithology as the rest of the sample. Texturally, pyrite occurs along the edge of some pyrrhotite grains (Fig. 2a); violarite occurs as individual grains and along the edge of a pyrrhotite grain (Fig. 2c,d). Two single grains of chalcopyrite were also identified (Fig. 2e). Representative compositional analyses are given in Table 2, and the major and minor element data for all sulfides analyzed are provided (Data Availability). Pyrrhotite grains are hexagonal, rectangular, lath shaped, or irregular (Fig. 2). The sample also contains carbonates, Ca-phosphate, and magnetite, and lacks anhydrous silicates.

### Sulfides in Ryugu particles A0094-01 and C0103-01

In comparison, Ryugu particles A0094-01 and C0103-01 (Fig. 1b, c) also contain abundant sulfides at 3.2 ± 0.1 and 3.4 ± 0.2 vol.%, respectively. The sulfides in A0094-01 are up to ~43 μm, and those in C0103-01 are up to ~92 μm in longest dimension (Fig. 1b, c). A0094-01 and C0103-01 both contain abundant Fe-depleted pyrrhotite, but in contrast to A0016, they contain more pentlandite grains (trace and 0.07 ± 0.01 vol.%, respectively) intergrown with pyrrhotite (Fig. 3a, c) but do not contain violarite, pyrite, or chalcopyrite. Pyrrhotite grains in both samples are hexagonal, rectangular, lath shaped, or less commonly irregular in shape (Fig. 3). Both A0094-01 and C0103-01 also contain carbonates, Ca-phosphates, and magnetite, and lack anhydrous silicates.

## Discussion

The violarite, pyrite, and chalcopyrite in A0016 are unlike anything previously observed in Ryugu and CI chondrites. Violarite is nearly unknown in astromaterials, except for that observed in the oxidized and hydrothermally altered type-4 Karoonda-like carbonaceous (CK) chondrite type sample, although it was not recognized as violarite at the time[19]. Violarite in Karoonda is highly Co-rich (up to 13.1 wt% Co) and intergrown with pentlandite. In comparison, violarite in Ryugu A0016 contains between 4.64 and 8.11 wt% Co, which is compositionally similar to Co-rich violarite from a terrestrial deposit that contains 0.19 to 9.69 wt% Co[26]. Pyrite is also exceptionally rare in astromaterials, and among chondritic material, it is only known in CK chondrites (data from[19]). The presence of pyrite indicates that A0016 was heavily oxidized during hydrothermal alteration, at an oxygen fugacity ($fO_2$) relative to the iron-wüstite (IW) buffer ≥ IW + 5.5 ($fO_2$ range from[20]). This $fO_2$ of alteration is higher than observed in CI chondrites, which were aqueously altered at an $fO_2$ between IW and IW + 5.5[20]. Chalcopyrite is rare in chondrites. It is present in CK and Rumuruti-like chondrites[19,20,27], and was recently reported in hydrothermally altered CI-like chondrites[21].

The presence of violarite, pyrite, and chalcopyrite in A0016 raises the question if this particle is exogenous to Ryugu or a sample of Ryugu with a distinct alteration history. Violarite and pyrite are present in CK chondrites, but the abundances and petrographic settings of sulfides and associated minerals in A0016 are unlike those in CK chondrites. For instance, pentlandite and pyrite are common but pyrrhotite is rare in CK chondrites[19,20]. In comparison, pyrrhotite is the dominant sulfide and pyrite and pentlandite are minor components in A0016 (Fig. 2). Also, A0016 lacks other mineral phases associated with CK chondrites, such as anhydrous silicate inclusions, Cr-bearing magnetite, or ilmenite[27,28], indicating it is not a fragment of a CK chondrite. Alternatively, this particle could be an exogeneous sample of an asteroid that collided with Ryugu but has not separately arrived to Earth as a meteorite or been recognized in our collections. However, based on (1) the morphological similarity of pyrrhotite to that in CI chondrites and other Ryugu particles (Figs. 2 and 3), (2) that the violarite, pyrite, and chalcopyrite in A0016 can be explained by the alteration of a CI chondrite precursor, and (3) the improbability of sampling an exogenous fragment that is so similar to other Ryugu particles and CI chondrites rather than something distinct, we infer that this particle is a unique sample native to Ryugu.

The compositions of pyrrhotite, pentlandite, pyrite, and violarite in A0016 agree with isothermal phase diagrams between 230 and

**Table 1 | Ryugu particle summary and presence of sulfide minerals**

| Sample | Area (mm²) | Sulfide abundance (vol% ± σ) | Pyrrhotite | Pentlandite | Violarite | Pyrite | Chalcopyrite | Sulfide Equilibration Temperature |
|---|---|---|---|---|---|---|---|---|
| A0016 | 5.93 | 4.1 ± 0.1 | ✓ | ✓ | ✓ | ✓ | ✓ | 230–400 °C |
| A0094-01 | 0.68 | 3.2 ± 0.1 | ✓ | ✓ | no | no | no | 25–135 °C |
| C0103-01 | 1.41 | 3.4 ± 0.2 | ✓ | ✓ | no | no | no | 25–135 °C |

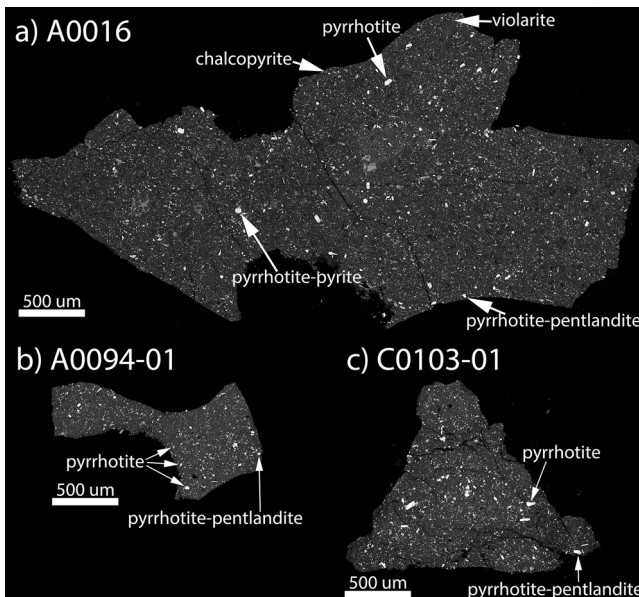

**Fig. 1 | Full backscattered electron images of samples, all at the same scale.**
**a** A0016 showing select location of sulfides and types (abundant pyrrhotite, with minor pyrite, violarite, and chalcopyrite; and a single pyrrhotite-pentlandite grain); **b** A0094-01 (pyrrhotite and pentlandite); and **c** C0103-01 (pyrrhotite and pentlandite). BSE images work by Z-contrast, so sulfides appear whitish while Mg-rich phyllosilicates appear dark gray.

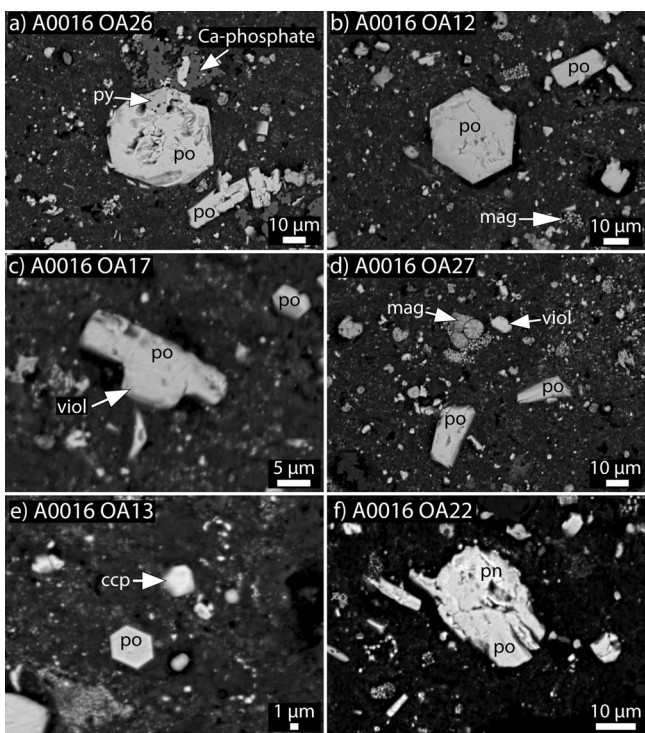

**Fig. 2 | Representative backscattered electron images of each sulfide mineral observed in A0016.** All sulfides are surrounded by a fine-grained phyllosilicate dominated matrix. **a** Roughly hexagonal grain of pyrrhotite (po) with pyrite (py), associated with Ca-phosphate. **b** Hexagonal and lath shaped pyrrhotite grains associated with magnetite framboids. **c** Violarite (viol) along the edge of pyrrhotite. **d** Violarite as a single grain, near magnetite, and grains of pyrrhotite. **e** Chalcopyrite (ccp) near a hexagonal grain of pyrrhotite. **f** Pyrrhotite-pentlandite (pn) intergrowth. OA opaque assemblage.

400 °C (Fig. 4a, b), indicating this sample of Ryugu was heated to temperatures at least within this range. The peak temperature could have been higher than 230–400 °C, as sulfide equilibration temperatures do not necessarily record peak thermal metamorphic temperatures of the host rock they reside within, but rather the temperature of sulfide equilibration during cooling from peak temperature[19]. This temperature range is supported by the formation conditions of violarite and chalcopyrite.

The presence of violarite as independent grains helps constrain their origins. In terrestrial settings and experiments, violarite was determined to form from the alteration of pentlandite[26,29–33], with direct experimental observations for pseudomorphic replacement[33]. Pentlandite occurs in some but not all CI chondrites either as independent grains in their matrices or in association with pyrrhotite along grain edges ([15,16,19]; A0094 and C0103 here, Fig. 3). The occurrence of violarite in A0016 as independent grains in the matrix, and along the edge of a pyrrhotite grain (Fig. 2c, d), is consistent with replacement of pentlandite by violarite. Moreover, the abundance of pentlandite in the unheated Ryugu sample C0103-01 (0.07 ± 0.01 vol.%) is similar to that of the violarite abundance in A0016 (0.07 ± 0.02 vol%). Therefore, we conclude that A0016 originally contained pentlandite that was then hydrothermally altered to violarite.

A minimum timescale for the replacement of pentlandite by violarite in A0016 can be constrained by laboratory experiments and terrestrial field observations, assuming that they are applicable to alteration on an asteroid. Laboratory experiments at acidic conditions (pH = 6 to 1) and mild hydrothermal conditions (80 to 210 °C) show that violarite formed from the pseudomorphic replacement of pentlandite between 33 and 87 days[33]. Violarite was also observed to form as an alteration product of freshly exposed pentlandite within 25 years in northern Ontario, Canada[29,30]. Given that the compositions of the violarite in A0016 indicate formation at temperatures between 230 and 400 °C (Fig. 4a, b), we find it more likely that the alteration timescale was on the order of months instead of years. Although alteration of CI chondrites reacting with the atmosphere was observed to alter sulfides to sulfates over time[3,34,35], violarite has never been observed in

a CI chondrite. Therefore, we do not find alteration of pentlandite to violarite in CI chondrites under terrestrial surface conditions (temperature, pH) to be likely. Mild acidic conditions (pH<6) to form violarite in A0016's parent body could have been achieved at temperatures between 230 and 400 °C via pyrrhotite oxidation releasing H⁺ into solution[36,37]. We conclude violarite formed in A0016 under hydrothermal conditions in Ryugu or its precursor parent body.

The presence of chalcopyrite in A0016, instead of cubanite, supports the temperature range of thermal alteration determined from pyrite, pentlandite, pyrrhotite, and violarite compositions (Fig. 4a, b). The orthorhombic form of cubanite is stable below 210 °C but transforms to isocubanite (high-temperature cubic form) when heated above 210 °C[38–41]. Upon cooling, isocubanite breaks down to chalcopyrite <210 °C[38,39,42]. Therefore, the presence of orthorhombic cubanite in CI chondrites[16] and previously studied Ryugu particles[3,22] indicates these samples were not heated above 210 °C. However, the presence of chalcopyrite instead of cubanite in Ryugu particle A0016 indicates that this sample was heated above 210 °C. As A0016 cooled, its original cubanite transformed to chalcopyrite below 210 °C.

In contrast to A0016, sulfide thermometry from pentlandite-pyrrhotite for both A0094-01 and C0103-01 indicate equilibration temperatures ≤100–135 °C, potentially as low as 25 °C (Fig. 4c, d). These temperatures are consistent with estimates for other Ryugu particles[3–9]. Due to the rarity and the sporadic occurrence of cubanite and sphalerite in CI chondrite sections[15,16], the lack of cubanite and sphalerite in A0094-01 and C0103-01 is not surprising due to their small sizes (Table 1). Although Ryugu particles A0016 and A0094 are both Chamber A particles from TD1, they are mineralogically distinct, and we conclude this is due to regolith mixing on Ryugu's surface. The lack of anhydrous silicates, and identification of carbonate, Ca-

**Table 2 | Representative sulfide analyses from Ryugu particle A0016**

| Particle | A0016 | A0016 | A0016 | A0016 | A0016 |
|---|---|---|---|---|---|
| Grain | OA34 | OA26 | OA6 | OA17 | OA22 |
| Analysis # | 88 | 54 | 102 | 42 | 114 |
| Mineral | Chalcopyrite | Fe-depleted Pyrrhotite | Pyrite | Violarite | Pentlandite |
| Chemical Composition (wt%) | | | | | |
| Fe | 30.65 | 59.17 | 44.71 | 26.16 | 29.06 |
| S | 34.78 | 39.01 | 50.90 | 41.54 | 33.02 |
| Si | 0.06 | 0.03 | 0.05 | 0.05 | 0.04 |
| Ni | 0.12 | 1.70 | 2.05 | 24.32 | 36.42 |
| Co | bdl | bdl | 1.70 | 6.97 | 0.47 |
| Cr | bdl | bdl | bdl | 0.05 | 0.06 |
| Cu | 32.50 | bdl | bdl | 0.06 | bdl |
| Total | 98.11 | 99.91 | 99.41 | 99.15 | 99.08 |
| Chemical Composition (at%) | | | | | |
| Fe | 25.53 | 45.94 | 32.63 | 20.36 | 23.85 |
| S | 50.48 | 52.76 | 64.70 | 56.32 | 47.22 |
| Si | 0.11 | 0.04 | 0.07 | 0.08 | 0.07 |
| Ni | 0.09 | 1.26 | 1.42 | 18.01 | 28.44 |
| Co | 0.00 | 0.00 | 1.17 | 5.14 | 0.36 |
| Cr | 0.00 | 0.00 | 0.00 | 0.04 | 0.05 |
| Cu | 23.79 | 0.00 | 0.00 | 0.04 | 0.00 |
| Total | 100.00 | 100.00 | 100.00 | 100.00 | 100.00 |
| Fe/S | 0.51 | 0.87 | 0.50 | 0.36 | 0.51 |
| Cations/S | 0.98 | 0.89 | 0.54 | 0.77 | 1.12 |
| bdl = below detection limit. | | | | | |
| P, Ti, Mn, and Mg are all bdl. | | | | | |
| Cations = Fe + Ni + Cr + Ti + Co + Cu. | | | | | |

Standards and detection limits in Supplementary Materials.
Analysis # = identifier of specific EPMA spot analysis, listed in Supplementary Information and Source Data.

phosphate, and magnetite in each sample studied here, indicate their high degree of aqueous alteration. Among aqueously altered carbonaceous chondrites,[20] showed with quantitative analyses that the at% Fe/S ratio of Ni-poor pyrrhotite (<1 wt% Ni) is an indicator of relative degree of aqueous alteration and oxidation, with lower at% Fe/S ratios indicating higher degrees of aqueous alteration and oxidation. The CI chondrites Orgueil, Alais, and Ivuna contain Ni-poor pyrrhotite with at % Fe/S ratios of 0.852 ± 0.007, 0.853 ± 0.008, and 0.869 ± 0.013[20]. The at% Fe/S ratio of Ni-poor pyrrhotite determined here for A0094-01 and C0103-01 are 0.860 ± 0.011 and 0.870 ± 0.009, respectively (±1σ), indicating they were aqueously altered to a comparable degree to the CI chondrites studied. No pyrrhotite grain analyzed in A0016 contains <1 wt% Ni, which may indicate either Ni in the pyrrhotite structure or the inclusion of pentlandite grains within pyrrhotite grains that are smaller than can be resolved via EPMA.

We infer that the precursor of A0016 was heavily aqueously altered CI chondrite material similar to other Ryugu particles, such as A0094 and C0103 studied here, that was then heated to at least 230–400 °C on a month(s)-long timescale. We note that dehydration and dehydroxylation of phyllosilicates occurs between 400 and 700 °C[43,44], and so the temperatures we estimate here were likely not high enough to dehydrate the phyllosilicates in Ryugu. However, the release of –OH/$H_2O$ from Fe-(oxy)hydroxides occurs between 200 and 400 °C, which likely occurred during the heating of A0016. If such material was plentiful on the surface of asteroid Ryugu, in addition to space weathering[24], it may help explain Hayabusa2's remote sensing observations of a weak 2.72 μm hydroxyl (-OH) band[10–12].

A comparison to the meteorite record provides additional, but distinct, evidence of hydrothermal alteration on the CI chondrite

parent body. Whereas typical CI chondrites record aqueous alteration temperatures below 100 °C[15,16] and contain sulfides similar to those in A0094 and C0103, some samples recently shown to be CI chondrites experienced thermal alteration[45–47]. For example, sulfides in the heated CI chondrite Yamato [Y]-86029[46] consist of troilite [FeS] and chalcopyrite[21] as a result of heating between 500 and 750 °C[13,48–50]. Such heating converted pre-existing Fe-depleted pyrrhotite and cubanite formed during earlier stages of aqueous alteration to troilite[20] and to chalcopyrite[21], respectively. In other words, the CI chondrite precursor parent body recorded a range of thermal alteration temperatures, potentially due to an impact that disrupted it[3,47].

The heating mechanism to form A0016 requires initial aqueous alteration to form CI-like material similar to A0094 and C0103 studied here, followed by hydrothermal alteration between 230 to 400 °C (Fig. 4a, b) for a timescale of a month to months to form violarite, pyrite, and chalcopyrite. The process that resulted in hydrothermally altering A0016 could have been: (1) radioactive decay in Ryugu's precursor parent body that led to hydrothermal alteration; (2) heating by close passage near the Sun; (3) and/or impact heating from an impact that disrupted the CI chondrite precursor parent body followed by burial and insulation.

Modeling radioactive heating of Ryugu's precursor body, assuming a ~ 50 km in radius[3], found that the interior would only reach a peak internal temperature of ~75 °C at ~5 million years after $t_0$. This peak internal temperature from radioactive decay is consistent with the initial aqueous alteration of A0016 to form CI-like sulfide minerals observed in other Ryugu particles (e.g., pyrrhotite, pentlandite, cubanite). However, we find radioactive decay an unlikely heat source to explain the later hydrothermal alteration between 230 and 400 °C.

Ryugu was proposed by[11] to have had a close passage near the Sun, during a different orbit than it has now. Solar radiation can heat material buried meters below the surface of an asteroid during close solar passage[12,51]. From Hayabusa2's remote sensing data, the strength and shape of the OH feature was found to indicate surface and subsurface heating above 300 °C[12], which is consistent with the temperature experienced by A0016. Heating via solar radiation at a range of perihelion distances were modeled, between 0.07 and 2 AU, with surface temperatures found to reach up to 777 °C[51]. However, the duration

of heating from solar radiation during close solar passages is dependent on the perihelion distance, and heating is expected to last only hours to days[51]. Such timescales are not long enough to melt water ice at and within meters of Ryugu's surface and keep it liquid between 230 and 400 °C for the month(s)-long timescales required to alter pentlandite to violarite in A0016. Therefore, we find solar heating of Ryugu's surface is unlikely to explain the hydrothermal alteration of A0016.

Impact heating of material that was previously aqueously altered and subsequent burial and insulation of this material is therefore the only explanation for the hydrothermal alteration of A0016. Temperatures as high as 700 °C could be achieved near the impact site of a 6-km radius impactor that that disrupted Ryugu's ~50 km radius precursor parent body, depending on the distance from the impact site[3]. The peak temperature of this range is much higher than that experienced by A0016, and so we infer that it was not directly in the impact site. In addition, A0016 does not contain shocked sulfides that would otherwise indicate high shock pressures (polycrystalline or fizzed sulfides[20,52]). Based on modeling by[3], temperatures between 300 and 500 °C would be reached between approximately 6–16 km from the impact site. Assuming the conditions of this impact[3] occurred on Ryugu's parent asteroid and considering A0016 records temperatures between 230 and 400 °C, A0016 could have been hydrothermally altered at the extent of this potential range from the impact site. Additional evidence for an impact history on Ryugu or its precursor parent body is supported by the presence of mildly shocked material discovered in Ryugu samples[53], as well as Lu-Hf isotope measurements indicating that some Ryugu particles record an impact-facilitated aqueous alteration event occurring over 1 billion years after radioactive heating ceased[14]. Therefore, we find it most likely that A0016 was near the site of a large impact on Ryugu's precursor parent body, perhaps the impact that disrupted it and led to the formation of Ryugu (Fig. 5). After impact, A0016 must have remained insulated, so that it experienced a stable fluid, likely acidic, on the timescale of a month to months at a temperature between 230 and 400 °C. This

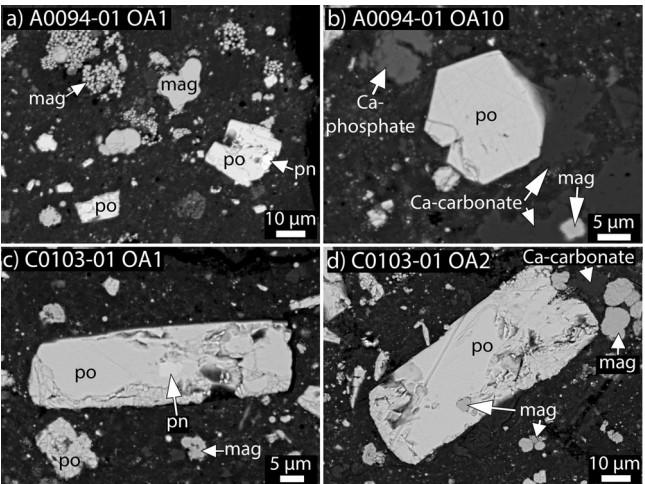

**Fig. 3 | Representative backscattered electron images of each sulfide mineral observed in A0094 and C0103.** All sulfides are surrounded by a fine-grained phyllosilicate dominated matrix. **a** Intergrowth of pyrrhotite (po) and pentlandite (pn), along with a grain of pyrrhotite and abundant magnetite (mag). **b** Hexagonal pyrrhotite grain associated with Ca-phosphate and magnetite. **c** Lath shaped pyrrhotite with interior pentlandite. **d** Lath shaped pyrrhotite with interior magnetite, with associated magnetite and Ca-carbonate. OA opaque assemblage.

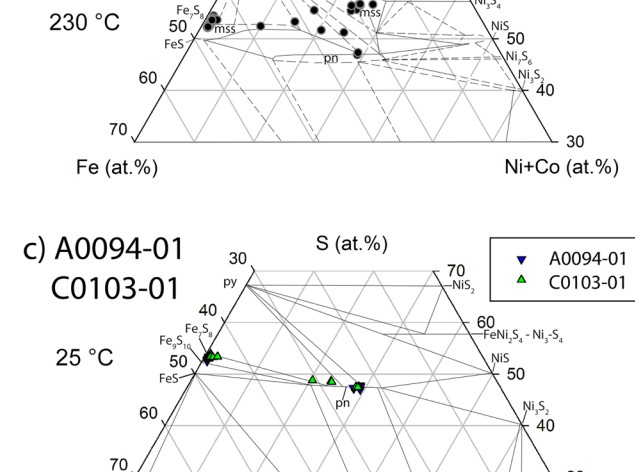

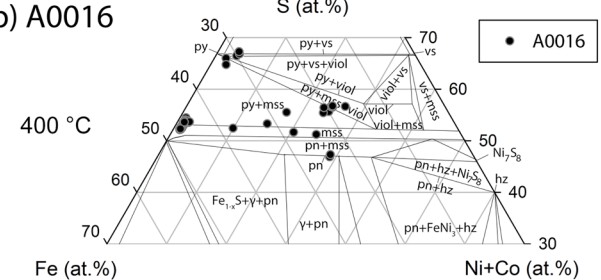

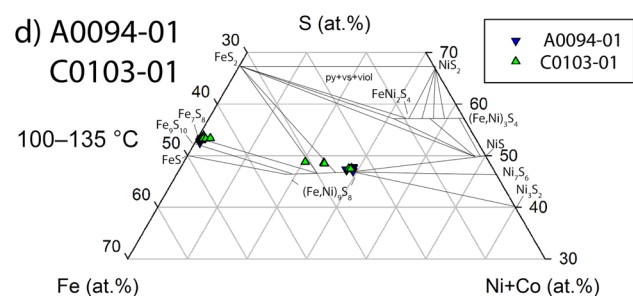

**Fig. 4 | In situ sulfide compositional data obtained via electron probe microanalyzer for each sample, superimposed on at% Fe-Ni + Co-S phase diagrams.** **a**, **b** A0016 and **c**, **d** A0094-01 and C0103-01. Phase diagrams at **a** 230 °C, **b** 400 °C, **c** 25 °C, and **d** 100–135 °C. Sulfides in A0016 are consistent with equilibrating between 230 and 400 °C, while A0094 and C0103 are consistent with equilibrating below 100–135 °C, potentially as low as 25 °C. Here po = pyrrhotite, pn =

pentlandite, py = pyrite, mss = monosulfide solid solution, hz = heazlewoodite (Ni$_3$S$_2$), vs = vaesite (NiS$_2$), viol = violarite (FeNi$_2$S$_4$), α = kamacite, and γ = taenite. Phase diagrams in **a**, **b** are adapted from[58], with original data for 230 °C from[59] and 400 °C from[60]. The 25 °C diagram is adapted from[61]. The 100 to 135 °C phase diagram is adapted from[62].

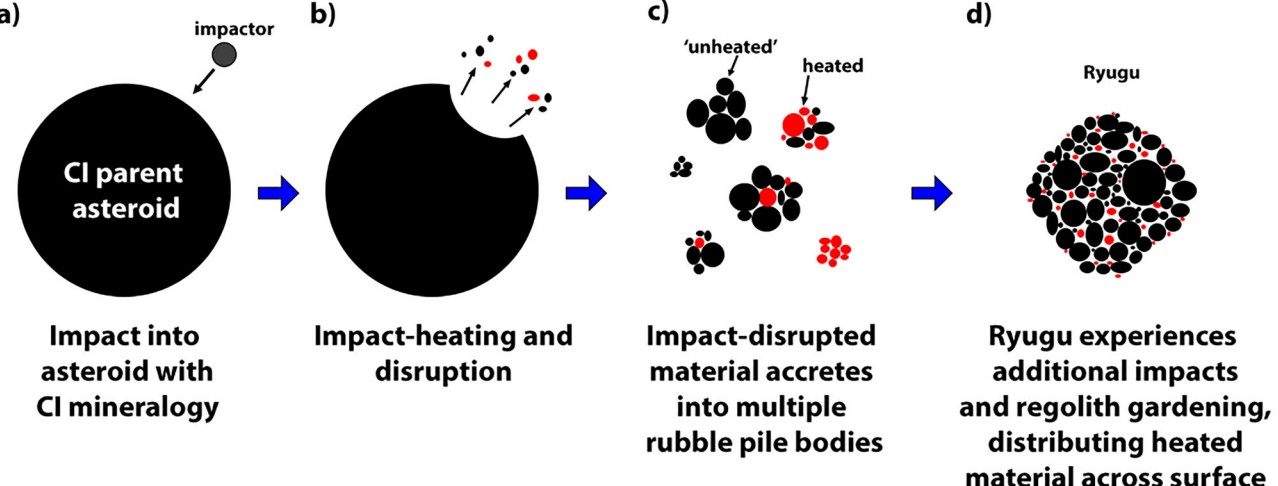

**Fig. 5 | Schematic diagram showing the progression of the impact-heating and disruption of the CI chondrite parent asteroid, to the accretion of Ryugu with unheated and heated material. a, b** Ivuna-like carbonaceous (CI) chondrite parent asteroid experiences aqueous alteration <100 °C. About 1 billion years ago, asteroid with CI chondrite mineralogy experiences a large disruptive impact, heating material up to 700 °C[3]. **c** Impact-disrupted material accretes into multiple CI chondrite rubble pile bodies, including bodies only of unheated material (i.e., <100 °C), mixtures of unheated and heated material, and those entirely composed of thermally altered CI material. **d** Ryugu accretes from mostly unheated material, but also accretes material heated by the impact. Buried and insulated heated material sustained temperatures between 230 and 400 °C on the order of a month, forming hydrothermally altered material like A0016. Regolith gardening later distributes this hydrothermally altered material across Ryugu, contributing to the weak 2.72 μm hydroxyl (-OH) band observed by Hayabusa2. Heated material is red. Unheated CI material is black. Sizes between stages **b** to **c**, and **c** to **d** are not to scale.

hydrothermal processing led to the transformation of pentlandite to violarite, the formation of pyrite, and the alteration of cubanite to chalcopyrite. Therefore, Ryugu particle A0016 and the heated CI chondrites provide evidence that their precursor parent body experienced hydrothermal alteration due a disruptive impact that led to the formation of Ryugu and other CI chondrite asteroids. A hydrothermally altered region, or regions, on a newly formed Ryugu may have later been disrupted by subsequent impacts, scattering this material across Ryugu's surface (Fig. 5). The presence of such hydrothermally altered material across Ryugu's surface, with contributions from space weathering and potentially heating from close solar approach[11,24], could explain the weak 2.72 μm hydroxyl (-OH) band observed by Hayabusa2.

## Methods
### Sample description and initial characterization
We studied sulfides in three separate Ryugu particles, A0016, A0094-01, and C0103-01. A0016 was received from JAXA as a single particle of 14.1 mg, with dimensions of 4.911 × 2.634 × 1.844 mm. At Arizona State University, A0016 particle was then mounted in epoxy and dry polished by hand following technique described in[54]. We noted during sample handling that the particle was not friable, it had sharp angles and was robust. The samples A0094-01 and C0103-01 were received as polished mounts, prepared by the method of[55] by JAXA. A0094 was originally a single particle of 1.8 mg, with dimensions of 2.417 × 1.063 mm. C0103 was originally a single particle of 1.5 mg, with dimensions of 1.975 × 1.182 mm.

Polished sections were initially characterized using an optical microscope to identify sulfides. We acquired X-ray element maps with 1 μm per pixel resolution, and high-resolution images with the JEOL-8530F Hyperprobe electron probe microanalyzer (EPMA) at Arizona State University (Figs. 1–3). Digital point counting using full grain backscattered electron (BSE) and composite X-ray element maps were used to determine the area of each particle (e.g., A0016, A0094-01, and C0103-01), and the abundance of sulfides and sulfide minerals in each particle (e.g.,[56]). Composite X-ray element maps of Fe, S, Ni, Co, Cu, Mg, Si, P, and Ca were used to identify individual minerals (Supplementary Information), which were verified either with energy-

dispersive X-ray spectroscopy (EDS) or with mineral analyses taken with the EPMA.

### Major and minor element compositional analysis
We determined the major and minor element compositions of sulfides in the Ryugu samples with the EPMA at ASU (Table 2 and Data Availability). Polished and carbon-coated polished sections were analyzed with a focused beam as individual points, with operating conditions of 15 kV and 20 nA, and a PAP correction method (a Phi-Rho-Z correction technique); peak and background counting times varied per element to optimize detection limits. Only sulfide analyses with totals between 97.5 and 102 wt% were retained. Standards and typical detection limits (wt%) are: troilite for Fe (0.05) and S (0.02); San Carlos olivine for Si (0.01) and Mg (0.01–0.02); schreibersite for P (0.01); rutile for Ti (0.02); rhodonite for Mn (0.04); NiS or Ni metal for Ni (0.03); Co metal for Co (0.02–0.03); Cr metal for Cr (0.04); and Cu metal for Cu (0.04). Minerals were identified based on in situ quantitative EPMA analyses, where major and minor element compositions, elemental ratios, and stoichiometry (Data Availability) were compared to known mineral formulas and compositions. All EPMA analyses, standards, and detection limits (for each analytical day) are included in the Source Data.

## Data availability
The data generated in this study have been deposited at https://doi.org/10.6084/m9.figshare.29525624[57]. The data generated in this study are also provided in the Supplementary Information and Source Data file. Source data are provided with this paper.

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

## Acknowledgements

We are thankful to JAXA for generously loaning the samples of A0016 (PI: T.J.Z.; CoI: D.L.S.), and A0094-01 and C0103-1 (PI: J.D.; CoI; D.L.S.), and to JAXA curation for their hard work curating and preparing these samples. We are also thankful for the hard work of the entire Hayabusa2 team. We are grateful to Axel Wittmann for assistance with the EPMA at ASU and Laurence Garvie for assistance with sample preparation. We acknowledge the use of facilities within the Eyring Materials Center at Arizona State University supported in part by NNCI-ECCS-1542160. This work was funded by NASA grant 80NSSC23K1268 (PI: T.J.Z.; CoI: D.L.S.).

## Author contributions

D.L.S. performed all analytical work and led data interpretation and writing of the manuscript. T.J.Z., M.B., and J.D. contributed to data interpretation and writing of the manuscript.

## Competing interests

The authors declare no competing interests.
