## [Transparent Peer Review file · Nature Communications]

Hydrothermal alteration of Ryugu from a disruptive impact recorded in a returned sample

Corresponding Author: Dr Devin Schrader

Version 0:

Reviewer comments:

Reviewer #1

(Remarks to the Author)

Review of „Hydrothermal alteration of Ryugu from a disruptive impact recorded in a returned sample” by D. L. Schrader et al.

Key result of this study is the discovery of certain sulfides (violarite, pyrite, chalcopyrite, pentlandite, Fe-depleted pyrrhotite) in a particle from asteroid (162173) Ryugu returned by JAXA's Hayabusa2 spacecraft (A0016), which have not been observed in any other Ryugu sample so far. From the presence of these specific sulfides and their compositions, the authors infer formation under hydrothermal conditions at 230 to 400 °C under highly oxidizing and potentially acidic conditions. This is in contrast to two other Ryugu particles from this study, and also to previously analyzed material, for which low-temperature alteration ($T \leq 100$ °C) under neutral to alkaline fluid conditions was concluded. The authors argue that the formation conditions of the sulfides found in A0016 are indicative of a large impact, possibly responsible for the disruption of Ryugu's precursor parent body.

This manuscript makes a significant contribution to our understanding of small asteroid formation and evolution in the nascent Solar System. The Ryugu samples constitute the first material from a C-type asteroid returned to Earth, and free from potential terrestrial contamination. As addressed in the paper, the majority of Ryugu material analyzed so far has been found to be CI-(chondrite-)like in composition, having experienced low-temperature alteration, whereas remote observations by Hayabusa2 showed indications of thermal alteration. From the discovery and study of violarite, pyrite, chalcopyrite, pentlandite, and Fe-depleted pyrrhotite in one specific Ryugu sample, the authors deduced a formation scenario involving a large impact on the asteroidal parent body. Alternative scenarios (material from the potential impactor so far unsampled in the meteoritic records; hydrothermal alteration by radioactive decay, heating of the precursor body during perihelion) are discussed by the authors.

The conclusions are adequately supported, and the alternative scenarios are sufficiently discussed (and refuted). Data analysis, interpretation and conclusions are presented coherently and comprehensibly. The methodology is sound, complying with quality standards. EPMA analysis is a well-established technique in the field, and the authors are demonstrably familiar and experienced with this kind of analysis. Sample designations, analytical parameters and sulfide grain positions are documented and given in the supplementary material; therefore, reproducibility of the results is ensured.

Questions/observations:

Since you extrapolate the formation conditions for violarite from terrestrial reactions and environment conditions, could the different gravitational settings (terrestrial vs. low-g/"nearly zero-g") have any influence on the sulfide formation or the estimated reaction temperatures? I am aware that this might be a quite tricky issue, but it could be interesting if you could give a short comment/sentence on the topic.

However, this is just a minor question; the manuscript makes a good and convincing case for the presented impact-driven hydrothermal alteration scenario responsible for the formation of the discovered sulfides. Thus, I recommend publication of this manuscript in Nature Communications.

One minor thing: From the context of the paragraph, I suppose in line 236, it should be "<1 at.%" instead of "<1 wt.%"?

Jan Leitner

Reviewer #2

(Remarks to the Author)

Manuscript Background:

The manuscript examines the results from elemental X-ray mapping of particles collected from asteroid Ryugu as part of the Jaxa Hayabusa 2 mission. Analysis is centred around improving our understanding of the dichotomy between spacecraft observation/measurement and the findings from analysis of returned particles which indicate aqueous alteration. The manuscript reports the results of analyses using x-ray mapping and EPMA of sulphides within the three grains studied. Results indicate that two grains are CI-chondrite like and have experienced aqueous alteration whilst one grain contains sulphide compositions indicating thermal alteration between 230-400 C. The study's conclusions are that impact heating of previously aqueously altered (CI-like) material is the only valid explanation for this occurrence and that the thermally altered grain must have been close to the impact site whilst within the Ryugu precursor.

This work is well written, well referenced and concise. It helps address a challenging scientific problem within the community using high-resolution analysis methods and robust data reporting.

Suggested Improvements:

- Within the introduction two possible scenarios are given to explain the disparity between the heated surface and aqueously altered inner material. It would be worthwhile revisiting these explanations in the discussion more explicitly.
- Similarly, it would be worthwhile referring to the two chambers (A & C) in the discussion as Chamber A (surface material) contains the thermally altered particle A0016, and the CI-like A0094. Are we to assume these have been juxtaposed by regolith mixing?
- How large are sulphides? Looking at Figure 1 there seems to be a significant difference in the sizes of the different sulphides. Whilst vol.% and morphology is provided and discussed brief statement on the sizes would perhaps be beneficial for completeness.
- Figure 2 referenced before figure 1

Reviewer #3

(Remarks to the Author)

Review of the paper "Hydrothermal alteration of Ryugu from a disruptive impact recorded in a returned sample" by Schrader et al. for Nature Communications

This manuscript presents new petrographic and chemical analyses of sulfides in three Ryugu particles, with particular emphasis on particle A0016. The identification of violarite, pyrite, and chalcopyrite provides compelling evidence for localized hydrothermal alteration at 230–400 °C, likely induced by an impact on Ryugu's precursor body. The work is novel, addresses an important discrepancy between spacecraft spectral observations (e.g., Kitazato et al.) and laboratory analyses (e.g., Ito et al., Yokoyama et al., T. Nakamura et al., E. Nakamura et al., Noguchi et al.), and is of broad significance for understanding asteroidal thermal and aqueous evolution.

Overall, the study presents important findings that merit publication in Nature Communications. With moderate revisions to strengthen the broader context, clarify some interpretations, and incorporate key references, the manuscript will be suitable for acceptance after another round of review.

My comments send separately.

REVIEWERS' COMMENTS

Reviewer #1 (Remarks to the Author):

Review of „Hydrothermal alteration of Ryugu from a disruptive impact recorded in a returned sample” by D. L. Schrader et al.

Key result of this study is the discovery of certain sulfides (violarite, pyrite, chalcopyrite, pentlandite, Fe-depleted pyrrhotite) in a particle from asteroid (162173) Ryugu returned by JAXA's Hayabusa2 spacecraft (A0016), which have not been observed in any other Ryugu sample so far. From the presence of these specific sulfides and their compositions, the authors infer formation under hydrothermal conditions at 230 to 400 °C under highly oxidizing and potentially acidic conditions. This is in contrast to two other Ryugu particles from this study, and also to previously analyzed material, for which low-temperature alteration ($T \leq 100$ °C) under neutral to alkaline fluid conditions was concluded. The authors argue that the formation conditions of the sulfides found in A0016 are indicative of a large impact, possibly responsible for the disruption of Ryugu's precursor parent body.

This manuscript makes a significant contribution to our understanding of small asteroid formation and evolution in the nascent Solar System. The Ryugu samples constitute the first material from a C-type asteroid returned to Earth, and free from potential terrestrial contamination. As addressed in the paper, the majority of Ryugu material analyzed so far has been found to be CI-(chondrite)-like in composition, having experienced low-temperature alteration, whereas remote observations by Hayabusa2 showed indications of thermal alteration. From the discovery and study of violarite, pyrite, chalcopyrite, pentlandite, and Fe-depleted pyrrhotite in one specific Ryugu sample, the authors deduced a formation scenario involving a large impact on the asteroidal parent body. Alternative scenarios (material from the potential impactor so far unsampled in the meteoritic records; hydrothermal alteration by radioactive decay, heating of the precursor body during perihelion) are discussed by the authors.

The conclusions are adequately supported, and the alternative scenarios are sufficiently discussed (and refuted). Data analysis, interpretation and conclusions are presented coherently and comprehensibly. The methodology is sound, complying with quality standards. EPMA analysis is a well-established technique in the field, and the authors are demonstrably familiar and experienced with this kind of analysis. Sample designations, analytical parameters and sulfide grain positions are documented and given in the supplementary material; therefore, reproducibility of the results is ensured.

Questions/observations:

Since you extrapolate the formation conditions for violarite from terrestrial reactions and environment conditions, could the different gravitational settings (terrestrial vs. low-g/”nearly zero-g”) have any influence on the sulfide formation or the estimated reaction temperatures? I am aware that this might be a quite tricky issue, but it could be interesting if you could give a short comment/sentence on the topic.

However, this is just a minor question; the manuscript makes a good and convincing case for the presented impact-driven hydrothermal alteration scenario responsible for

the formation of the discovered sulfides. Thus, I recommend publication of this manuscript in Nature Communications.

- Thank you for your thoughtful observation. You are right that gravitational differences between terrestrial and low-g environments could, in principle, influence fluid transport efficiency and therefore sulfide formation conditions. However, given that the sulfide formation in our scenario is driven primarily by localized, impact-induced hydrothermal systems—where temperature and fluid-rock interactions dominate—the influence of gravity is likely secondary. In addition, comparison between terrestrial experiments at 1 bar (atmosphere) and asteroidal formation conditions are common in the field. Nonetheless, we've added a brief note acknowledging our assumption in the revised discussion.

On Page 8 we have added, "A minimum timescale for the replacement of pentlandite by violarite in A0016 can be constrained by laboratory experiments and terrestrial field observations, assuming that they are applicable to alteration on an asteroid."

One minor thing: From the context of the paragraph, I suppose in line 236, it should be "<1 at.%" instead of "<1 wt.%"?

-Thank you for catching this. It should be <1 wt.% Ni on Line 236, but Line 230 should not have been at.%. Line 230 has been corrected to wt.%.

Jan Leitner

Reviewer #2 (Remarks to the Author):

Manuscript Background:

The manuscript examines the results from elemental X-ray mapping of particles collected from asteroid Ryugu as part of the Jaxa Hayabusa 2 mission. Analysis is centred around improving our understanding of the dichotomy between spacecraft observation/measurement and the findings from analysis of returned particles which indicate aqueous alteration. The manuscript reports the results of analyses using x-ray mapping and EPMA of sulphides within the three grains studied. Results indicate that two grains are CI -chondrite like and have experienced aqueous alteration whilst one grain contains sulphide compositions indicating thermal alteration between 230-400 C. The study's conclusions are that impact heating of previously aqueously altered (CI-like) material is the only valid explanation for this occurrence and that the thermally altered grain must have been close to the impact site whilst within the Ryugu precursor.

This work is well written, well referenced and concise. It helps address a challenging scientific problem within the community using high-resolution analysis methods and robust data reporting.

Suggested Improvements:

- Within the introduction two possible scenarios are given to explain the disparity between the heated surface and aqueously altered inner material. It would be worthwhile revisiting these explanations in the discussion more explicitly.

- We appreciate your suggestion to revisit the two scenarios in the discussion. These scenarios are included in the Discussion section, with heating via close solar approach receiving a detailed discussion on page 12. In addition, on page 14 we have added new text to clarify that space weathering is likely still a contributor to the remote sensing data.

“The presence of such hydrothermally altered material across Ryugu’s surface, with contributions from space weathering and potentially heating from close solar approach [11,24], could explain the weak 2.72 μm hydroxyl (-OH) band observed by Hayabusa2.”

- Similarly, it would be worthwhile referring to the two chambers (A & C) in the discussion as Chamber A (surface material) contains the thermally altered particle A0016, and the CI-like A0094. Are we to assume these have been juxtaposed by regolith mixing?

-The particles are introduced as Chamber A (A0016 and A0094-01) and Chamber C (C0103-01) in the results section. To avoid redundancy, we chose to not refer to them as such throughout the manuscript. However, we do agree clearly stating that A0016 and A0094 are from the same collection site, despite being different from one another, improves the clarity of the discussion.

We have added clarification text on Page 10 to state that A0016 and A0094, while both from Chamber A (TD1), they are mineralogically distinct, which supports regolith mixing.

“While Ryugu particles A0016 and A0094 are both Chamber A particles from TD1, they are mineralogically distinct, and we conclude this is due to regolith mixing on Ryugu’s surface.”

- How large are sulphides? Looking at Figure 1 there seems to be a significant difference in the sizes of the different sulphides. Whilst vol.% and morphology is provided and discussed brief statement on the sizes would perhaps be beneficial for completeness.

-We agree this is a good parameter to constrain and include. We have added the maximum sizes of sulfides in each sample to the Results section for all three samples. One Page 5: “The sulfides are up to ~70 μm in longest dimension (Fig. 1a).”
On Page 6: “The sulfides in A0094-01 are up to ~43 μm , and those in C0103-01 are up to ~92 μm in longest dimension (Fig. 1b,c).”

- Figure 2 referenced before figure 1

-Thank you for catching this, on Page 5 we have corrected the original incorrect reference of just Figure 2 to referencing both Figure 1a and 2 (both are for A0016).

Reviewer #3 (Remarks to the Author):

Review of the paper “Hydrothermal alteration of Ryugu from a disruptive impact recorded in a returned sample” by Schrader et al. for Nature Communications

This manuscript presents new petrographic and chemical analyses of sulfides in three Ryugu particles, with particular emphasis on particle A0016. The identification of violarite, pyrite, and chalcopyrite provides compelling evidence for localized hydrothermal alteration at 230–400 °C, likely induced by an impact on Ryugu’s precursor body. The work is novel, addresses an important discrepancy between spacecraft spectral observations (e.g., Kitazato et al.) and laboratory analyses (e.g., Ito et al., Yokoyama et al., T. Nakamura et al., E. Nakamura et al., Noguchi et al.), and is of broad significance for understanding asteroidal thermal and aqueous evolution. Overall, the study presents important findings that merit publication in Nature Communications. With moderate revisions to strengthen the broader context, clarify some interpretations, and incorporate key references, the manuscript will be suitable for acceptance after another round of review.

Major Points

Significance

- The manuscript could more explicitly connect its findings to the broader debate on the thermal and aqueous histories of CI-like bodies, including implications for volatile retention, redox evolution, and the apparent discrepancy between surface spectral features and bulk sample analyses.

-We agree that the broader context of CI-like parent body evolution is important. In the current manuscript, we address this connection by comparing A0016 to heated CI-like meteorites (see Page 11; the discussion of Y-86029 chondrite) and infer what this means for the CI chondrite precursor body, which we believe provides a relevant link to ongoing discussions about thermal processing and volatile behavior in these materials. While a more detailed treatment of CI chondrite evolution, including redox trends, volatile retention, detailed spectroscopic comparisons, is beyond the scope of this study, we acknowledge that our findings contribute to this broader field and may help inform future work focused specifically on these questions.

- A brief comparison to asteroid Bennu (OSIRIS-REx), which also exhibits CI-like characteristics, would strengthen the discussion and highlight the general relevance of the results.

-Agreed. On Page 3 to 4 we have included a reference to a recent paper (Zega et al., 2025; <https://doi.org/10.1038/s41561-025-01741-0>) that found sulfide equilibration temperatures of 25 °C, similar to that of typical Ryugu samples. “Recent analyses of samples returned from asteroid Bennu are noted to be similar to both Ryugu and the CI-chondrites, and have also undergone low-temperature aqueous alteration as low as 25 °C (e.g., [18]).”

- While the authors conclude that A0016 is indigenous to Ryugu, the discussion of possible exogenous origins remains relatively brief. Expanding this section with more

quantitative constraints (e.g., the statistical likelihood of sampling an exogenous fragment, or isotopic evidence that rules this out) would make the argument more robust.

- Thank you for raising this important point. While we acknowledge that the discussion of exogenous origins for A0016 could be clarified, we have based our conclusion that A0016 is indigenous to Ryugu on several lines of evidence, including textural and mineralogical characteristics, which are consistent with mild hydrothermal alteration of typical Ryugu particles. Although we do not have isotopic data to further constrain its origin, we have expanded the discussion to more clearly articulate the reasoning behind our interpretation. Specifically, we note that the likelihood of sampling an exogenous fragment with such close textural and mineralogical similarities (except the identified distinct sulfides) to other Ryugu samples is low, particularly given the size of the returned mass and the consistency of A0016 with known CI lithologies. We have added a statement to the manuscript to clarify this point (below). In addition, since this particle is so CI-like, isotopic analyses typically used for this purpose (Cr, Ti, O, H, C, N) may not provide any further clarity as to its origin as the CI chondrites are known to have a wide range of isotopic compositions (e.g., see Schrader et al., 2025; <https://doi.org/10.1016/j.gca.2024.12.021>). In addition, the sample was not allocated or approved for such analyses by JAXA, the grain size was too small to facilitate many of those bulk isotopic analyses to the accuracy needed and are now not possible due to the sample being mounted in epoxy. We present our evidence clearly and provide the best conclusion possible based on available evidence.

Importantly, if A0016 is exogeneous, it is from a CI-like chondrite impactor, and the presence of such heated material on Ryugu still provides a solution to the remote sensing discrepancy at the core of our work.

We have reworded the section addressing this issue and added new text for clarity on Page 7.

“Also, A0016 lacks other mineral phases associated with CK chondrites, such as anhydrous silicate inclusions, Cr-bearing magnetite, or ilmenite [27,28], indicating it is not a fragment of a CK chondrite. Alternatively, this particle could be an exogeneous sample of an asteroid that collided with Ryugu but has not separately arrived to Earth as a meteorite or been recognized in our collections. However, based on (1) the morphological similarity of pyrrhotite to that in CI chondrites and other Ryugu particles (**Figs. 2 and 3**), (2) that the violarite, pyrite, and chalcopyrite in A0016 can be explained by the alteration of a CI chondrite precursor, and (3) the improbability of sampling an exogenous fragment that is so similar to other Ryugu particles and CI chondrites rather than something distinct, we infer that this particle is a unique sample native to Ryugu. ”

Alteration timescales

- The inference that pentlandite-to-violarite conversion occurred on month-scale timescales is intriguing but should be treated with caution. The manuscript would benefit from a more explicit discussion of the uncertainties involved in extrapolating laboratory kinetic data to parent-body conditions, particularly regarding whether such transient

events could realistically be preserved in Ryugu's long-term thermal history. In addition, the temperature constraints on violarite formation require clarification. Previous experimental studies have shown that violarite can form at lower temperatures than the 230–400 °C range cited here.

This discrepancy could be addressed by contrasting two scenarios: (1) equilibrium assemblages reflecting peak metamorphic temperatures, or (2) re-equilibration during postimpact hydrothermal cooling, when conditions may have persisted long enough (months?) to allow the pentlandite-to-violarite transformation. Clarifying these points would strengthen the interpretation.

-Agreed, violarite can form at lower temperatures (between 80 and 210 °C, taking between 33–87 days), as stated on Page 8. However, we made the argument for higher temperature formation of violarite and thus the shorter month-long-timescale because (1) the composition of violarite is consistent with sulfide equilibration between 230–400 °C (from phase diagrams shown in Fig. 4a,b), (2) low-temperature formation requires much longer timescales, such as the 25 years stated for violarite for form on Earth's surface in Canada, and (3) the high temperature formation is in agreement with that required to form chalcopyrite. If the violarite in A0016 formed at 'low' temperature, its composition would not agree with the 230–400 °C phase diagrams. We have reworded the sentence on page 9 for clarity that the violarite composition is what indicates this higher temperature, and thus shorter timescale for formation.

On page 9: "Given that the compositions of the violarite in A0016 indicate formation at temperatures between 230 to 400°C (**Fig. 4a,b**), we find it more likely the alteration timescale was on the order of months instead of years."

Link to remote sensing observations

- The proposed connection between A0016's hydrothermal alteration and the weak 2.7 μm OH absorption observed in spacecraft spectra is intriguing. This point could be expanded to address whether such localized hydrothermal regions might be widespread on Ryugu's surface, and how this affects interpretations of global remote sensing data.

- We agree that the potential connection between hydrothermal alteration, such as that recorded in A0016, and the weak 2.7 μm OH absorption observed in spacecraft spectra warrants further discussion in the manuscript. As noted in the manuscript, the estimated heating conditions for A0016 (230–400 °C) are sufficient to release –OH/H₂ O from Fe-(oxy)hydroxides without fully dehydrating phyllosilicates. We have included new text to the end of the manuscript to make the connection to spacecraft observations and this material clearer, including a reference to the new schematic diagram suggested by this reviewer.

On page 14: "These hydrothermally altered region, or regions, on a newly formed Ryugu may have later been disrupted by subsequent impacts, scattering this material across Ryugu's surface (**Fig. 5**). The presence of such hydrothermally altered material across Ryugu's surface, with contributions from space weathering and potentially heating from close solar approach [11,24], could explain the weak 2.72 μm hydroxyl (-OH) band observed by Hayabusa2."

References to strengthen context

The manuscript would benefit from recent Nature Astronomy papers that Ryugu's alteration history.

-Agreed, all of these references have now been included in the manuscript.

- Ito et al. (2022) should be cited in sections comparing Ryugu with CI chondrites, particularly regarding low-temperature (<30 C) alteration inferred from organics in phyllosilicates.

Added citation to the first paragraph of the Introduction on Page 2, which includes multiple studies that showed the alteration was at low-temperatures.

Ito, M., et al. A pristine record of outer Solar System materials from asteroid Ryugu's returned sample. *Nature Astronomy* 6: 1163–1171 (2022).

- McCain et al. (2023) should be cited in the introduction, as their study of O isotopes in calcite indicated similar low alteration temperatures and also constrained the precursor body size to <20 km in diameter.

Added citation to the first paragraph of the Introduction on Page 2, which includes multiple studies that showed the alteration was at low-temperatures.

McCain, K. A. et al. Early fluid activity on Ryugu inferred by isotopic analyses of carbonates and magnetite. *Nature Astronomy* 7: 309–317 (2023).

From McCain et al. (2023)'s abstract "Carbonate ages show that this fluid–rock interaction took place within approximately the first 1.8 million years of Solar System history, requiring early accretion either in a planetesimal less than ~20 km in diameter or within a larger body that was disrupted and reassembled." Their work also supports disruption of a larger body. In addition, we clearly state that we work from the model of Nakamura et al. (2023) for the diameter of the precursor body.

- Yamaguchi et al. (2023) should be cited when discussing sulfides, as their study documents sulfide diversity and redox implications in Ryugu samples.

Added to introduction, page 2 (about temperature) and page 4 (sulfide mineralogy).

Yamaguchi, A., et al. Insight into multi-step geological evolution of C-type asteroids from Ryugu particles. *Nature Astronomy* 7: 398–405 (2023).

- Tomioka et al. (2023) could be incorporated in the discussion, since their identification of mild-shock features in Ryugu particles implies transient heating and oxidation, providing context for the present interpretation.

Added to discussion, page 13, to support evidence for impact.

[53] Tomioka, N., et al. A history of mild shocks experienced by the regolith particles on hydrated asteroid Ryugu. *Nature Astronomy* 7: 669–677 (2023).

“Additional evidence for an impact history on Ryugu or its precursor parent body is supported by the presence of mildly shocked material discovered in Ryugu samples [53], as well as Lu-Hf isotope measurements indicating that some Ryugu particles record an impact-facilitated aqueous alteration event occurring over 1 billion years after radioactive heating ceased [14].”

Incorporating these citations would situate the present study more firmly within the current state of knowledge.

Minor Points

1. Abstract: It would be useful to highlight more explicitly that this is the first direct evidence for hydrothermal alteration at 230–400 °C in Ryugu.

-Change made to Abstract. “The presence and compositions of violarite, pyrite, chalcopyrite, pentlandite, and Fe-depleted pyrrhotite grains provide the first direct evidence for hydrothermal alteration between 230 to 400 °C under highly oxidizing and potentially acidic fluid conditions.”

2. Line 157: Please clarify how fO_2 estimates were derived from the sulfide assemblages (e.g., which equilibria or calibration were applied).

-Thank you for the comment. The reported fO_2 ranges reflect conditions typical of chondritic materials where these sulfide assemblages are known to occur, as discussed in Schrader et al. (2021, GCA) (i.e., ref 20).

This has now been clarified in the text on page 7 as “(fO_2 range from [20])”.

Additionally, fO_2 values for these samples were originally calculated by Righter and Neff (2007) using equilibria involving magnetite–ilmenite, quartz–iron–fayalite, and quartz–iron–ferrosilite buffer reactions. The relevance and applicability of these redox conditions to the sulfide assemblages are thoroughly discussed in Schrader et al. (2021, GCA, 303, 66–91), which is cited in the manuscript.

3. Methods: Please include a brief summary of detection limits and phase-identification criteria (currently in Supplementary) in the main text, for accessibility to a broader readership.

-Added to Methods on Page 15: “Standards and typical detection limits (wt.%) are: troilite for Fe (0.05) and S (0.02); San Carlos olivine for Si (0.01) and Mg (0.01–0.02); schreibersite for P (0.01); rutile for Ti (0.02); rhodonite for Mn (0.04); NiS or Ni metal for Ni (0.03); Co metal for Co (0.02–0.03); Cr metal for Cr (0.04); and Cu metal for Cu (0.04). Minerals were identified based on in situ quantitative EPMA analyses, where major and minor element compositions, elemental ratios, and stoichiometry (Data Availability) were compared to known mineral formulas and compositions. All EPMA analyses, standards, and detection limits (for each analytical day) are included in the Source Data.”

4. Additional figure: A schematic illustration summarizing the proposed alteration sequence (primary aqueous alteration -> impact heating -> hydrothermal alteration -> reaccrution) would improve clarity. The style of Fig. 5 in Yamaguchi et al. (2023) would be useful.

-Thank you for this suggestion. We have included a schematic diagram (new Figure 5) to more clearly convey our formation scenario and referenced it in the Discussion.

Review of the paper “Hydrothermal alteration of Ryugu from a disruptive impact recorded in a returned sample” by Schrader et al. for *Nature Communications*

This manuscript presents new petrographic and chemical analyses of sulfides in three Ryugu particles, with particular emphasis on particle A0016. The identification of violarite, pyrite, and chalcopyrite provides compelling evidence for localized hydrothermal alteration at 230–400 °C, likely induced by an impact on Ryugu’s precursor body. The work is novel, addresses an important discrepancy between spacecraft spectral observations (e.g., Kitazato et al.) and laboratory analyses (e.g., Ito et al., Yokoyama et al., T. Nakamura et al., E. Nakamura et al., Noguchi et al.), and is of broad significance for understanding asteroidal thermal and aqueous evolution.

Overall, the study presents important findings that merit publication in *Nature Communications*. With moderate revisions to strengthen the broader context, clarify some interpretations, and incorporate key references, the manuscript will be suitable for acceptance after another round of review.

Major Points

Significance

- The manuscript could more explicitly connect its findings to the broader debate on the thermal and aqueous histories of CI-like bodies, including implications for volatile retention, redox evolution, and the apparent discrepancy between surface spectral features and bulk sample analyses.
- A brief comparison to asteroid Bennu (OSIRIS-REx), which also exhibits CI-like characteristics, would strengthen the discussion and highlight the general relevance of the results.
- While the authors conclude that A0016 is indigenous to Ryugu, the discussion of possible exogenous origins remains relatively brief. Expanding this section with more quantitative constraints (e.g., the statistical likelihood of sampling an exogenous fragment, or isotopic evidence that rules this out) would make the argument more robust.

Alteration timescales

- The inference that pentlandite-to-violarite conversion occurred on month-scale timescales is intriguing but should be treated with caution. The manuscript would benefit from a more explicit discussion of the uncertainties involved in extrapolating laboratory kinetic data to parent-body conditions, particularly regarding whether such transient events could realistically be preserved in Ryugu’s long-term thermal history. In addition, the temperature constraints on violarite formation require clarification. Previous experimental studies have shown that violarite can form at lower temperatures than the 230–400 °C range cited here.

This discrepancy could be addressed by contrasting two scenarios: (1) equilibrium assemblages reflecting peak metamorphic temperatures, or (2) re-equilibration during post-impact hydrothermal cooling, when conditions may have persisted long enough (months?) to allow the pentlandite-to-violarite transformation. Clarifying these points would strengthen the interpretation.

Link to remote sensing observations

- The proposed connection between A0016's hydrothermal alteration and the weak 2.7 μm OH absorption observed in spacecraft spectra is intriguing. This point could be expanded to address whether such localized hydrothermal regions might be widespread on Ryugu's surface, and how this affects interpretations of global remote sensing data.

References to strengthen context

The manuscript would benefit from recent Nature Astronomy papers that Ryugu's alteration history.

- Ito et al. (2022) should be cited in sections comparing Ryugu with CI chondrites, particularly regarding low-temperature (<30 °C) alteration inferred from organics in phyllosilicates.
- McCain et al. (2023) should be cited in the introduction, as their study of O isotopes in calcite indicated similar low alteration temperatures and also constrained the precursor body size to <20 km in diameter.
- Yamaguchi et al. (2023) should be cited when discussing sulfides, as their study documents sulfide diversity and redox implications in Ryugu samples.
- Tomioka et al. (2023) could be incorporated in the discussion, since their identification of mild-shock features in Ryugu particles implies transient heating and oxidation, providing context for the present interpretation.

Incorporating these citations would situate the present study more firmly within the current state of knowledge.

Minor Points

1. **Abstract:** It would be useful to highlight more explicitly that this is the *first direct evidence for hydrothermal alteration at 230–400 °C in Ryugu*.
2. **Line 157:** Please clarify how $f\text{O}_2$ estimates were derived from the sulfide assemblages (e.g., which equilibria or calibration were applied).
3. **Methods:** Please include a brief summary of detection limits and phase-identification criteria (currently in Supplementary) in the main text, for accessibility to a broader readership.

4. **Additional figure:** A schematic illustration summarizing the proposed alteration sequence (primary aqueous alteration -> impact heating -> hydrothermal alteration -> reaccrction) would improve clarity. The style of Fig. 5 in Yamaguchi et al. (2023) would be useful.